# Distinct Effects of Cannabidiol on Sphingolipid Metabolism in Subcutaneous and Visceral Adipose Tissues Derived from High-Fat-Diet-Fed Male Wistar Rats

**DOI:** 10.3390/ijms23105382

**Published:** 2022-05-11

**Authors:** Klaudia Berk, Karolina Konstantynowicz-Nowicka, Tomasz Charytoniuk, Ewa Harasim-Symbor, Adrian Chabowski

**Affiliations:** Department of Physiology, Medical University of Bialystok, 15-089 Bialystok, Poland; karolina.konstantynowicz@umb.edu.pl (K.K.-N.); tomasz.charytoniuk@umb.edu.pl (T.C.); eharasim@umb.edu.pl (E.H.-S.); adrian@umb.edu.pl (A.C.)

**Keywords:** cannabidiol, insulin resistance, sphingolipids, adipose tissue, ceramide

## Abstract

Available data suggest that cannabidiol (CBD) may ameliorate symptoms of insulin resistance by modulating the sphingolipid concentrations in particular organs. However, it is not entirely clear whether its beneficial actions also involve adipose tissues in a state of overnutrition. The aim of the study was to evaluate the effect of CBD on sphingolipid metabolism pathways and, as a result, on the development of insulin resistance in subcutaneous (SAT) and visceral (VAT) adipose tissues of an animal model of HFD-induced insulin resistance. Our experiment was performed on Wistar rats that were fed with a high-fat diet and/or received intraperitoneal CBD injections. We showed that CBD significantly lowered the ceramide content in VAT by reducing its de novo synthesis and increasing its catabolism. However, in SAT, CBD decreased the ceramide level through the inhibition of salvage and de novo synthesis pathways. All of these changes restored adipose tissues’ sensitivity to insulin. Our study showed that CBD sensitized adipose tissue to insulin by influencing the metabolism of sphingolipids under the conditions of increased availability of fatty acids in the diet. Therefore, we believe that CBD use may be considered as a potential therapeutic strategy for treating or reducing insulin resistance, T2DM, and metabolic syndrome.

## 1. Introduction

In the 21st century, obesity and type 2 diabetes mellitus (T2DM) have become a global health problem, and the need for the development of novel and safer therapeutic interventions is great. Obesity-related comorbidities generate high costs for health systems and diminish patients’ quality of life. One of these comorbidities is insulin resistance (IR), which is a pathophysiological state where vital metabolic organs (liver, muscle, adipose tissues) demonstrate reduced sensitivity to the activity of insulin [1,2,3]. IR may be distinguished as a risk factor for the occurrence of T2DM together with chronic inflammation and excessive visceral fat deposition [4].

Human adipose tissue is commonly divided, according to the regional fat distribution, into subcutaneous (SAT) and visceral adipose tissue (VAT) [5]. These two classes of fat deposits have intrinsically distinct gene expression profiles and different lipid metabolisms [6,7]. VAT exerts higher lipolytic activity and lower insulin sensitivity than SAT [8]. The accumulation of lipids in VAT impairs insulin sensitivity in metabolically obese, normal-weight individuals [9]. On the other hand, preferential deposition of fat in subcutaneous deposits, such as those detected in most women, positively correlates with metabolically healthy obesity. It is a condition in which, despite the occurrence of obesity, individuals may be protected from T2DM and the development of cardiovascular diseases [10].

Undoubtedly, adipocytes’ insulin sensitivity indirectly regulates liver and muscle function by modulating systemic glucose and fatty acid homeostasis [11]. A large lipid pool in adipose tissue caused by excessive non-esterified fatty acids (NEFAs) influx provokes the hypertrophy of adipocytes and a hyper-lipolytic phenotype that is resistant to insulin [12]. Furthermore, the disruption of lipid metabolism pathways results in elevated plasma levels of triacylglycerols (TAGs) [13], ceramide (CER) [14], and sphingosino-1-phosphate (S1P) [15] in obese individuals [13]. CER and S1P are sphingolipids that are key bioactive signaling molecules in adipose tissue [16]. Altered sphingolipid metabolism induces a dysfunctional adipose tissue phenotype by blocking insulin signaling and adipogenesis and promoting chronic inflammation [17]. Thus, a new therapeutic strategy focused on the modulation of sphingolipid metabolism is needed in order to restore the regular function and insulin sensitivity of adipose tissue [18].

The lack of an effective insulin resistance treatment has led to extensive pharmacological and physiological research, particularly on non-euphoric phytocannabinoids. One of them is CBD, which is found in medicinal and fiber-type Cannabis sativa plants (hemp) and affects lipid metabolism, but, unlike Δ^9^-THC, it lacks abuse potential [19,20]. Many research works have proved that CBD modulates both lipid and glucose homeostasis by targeting various receptors, as well as regulating endocannabinoid levels [21,22,23]. A growing body of studies suggest that CBD may be a pharmacological tool in ameliorating symptoms of insulin resistance, T2DM, and metabolic syndrome [20]. Now, we know that CBD reduces diabetes incidence in non-obese diabetic mice, has beneficial effects on complications from diabetes, and prevents palmitate-induced insulin sensitivity impairment when differentiating bone-marrow-derived mesenchymal stem cells by recovering Akt activation and increasing GLUT4 mRNA expression [24,25]. Nonetheless, there are still no data describing the possible impacts of CBD on insulin signaling in the two deposits of adipose tissue under normal and obese conditions. Similarly, it is unknown if CBD affects the metabolism of sphingolipids in adipocytes. Thus, we performed our study to delineate the novel effect of CBD on sphingolipid metabolism pathways and, as a result, on the development of insulin resistance in subcutaneous and visceral adipose tissue.

## 2. Results

### 2.1. Influence of CBD on the Intracellular Sphingolipid Concentration in SAT and VAT of Rats Fed with Standard and High-Fat Diets

In subcutaneous adipose tissue, we noticed substantial increases in SFO, SFA, S1P, and ceramide in the HFD group (+106.9%; +36.2%; +72.4%; +41.5%; *p* < 0.05; Figure 1A,B,C, respectively) and no changes in the SFA1P level (−19.9%; *p* > 0.05; Figure 1D).

In the group of rats injected with CBD, the level of SFO was significantly elevated (+66.4%; *p* < 0.05; Figure 1A) compared to the rats fed with the standard diet. Moreover, in this group, the concentrations of other sphingolipids remained unchanged (SFA: −6%; S1P: +1%; SFA1P: −24.3%; CER: −6.7% *p* > 0.05; Figure 1B,D,E, respectively).

After the simultaneous administration of HFD and CBD, we observed a substantial rise in the SFO and S1P levels compared to the control animals (+47.0%, +35.8%; *p* < 0.05; Figure 1A,C) and a significant decline in the SFO, SFA, and CER levels compared to the HFD-fed subjects (−59.9%, −41.6%, −48.8%; *p* < 0.05; Figure 1A,B,E, respectively). The level of SFA1P did not change in the HFD+CBD group (Figure 1D).

In visceral adipose tissue, the high-fat diet caused a substantial reduction in the SFO level (−152.7%; *p* < 0.05; Figure 1A) and a marked increase in SFA, S1P, SFA1P, and CER (+70.6%; +200.4%; +43.9%; + 32.6%; *p* < 0.05; Figure 1B,C,D,E, respectively). In the group treated with CBD, we showed a significant decrease in the SFO and SFA levels compared to those in the control group (−100.4%; −43.4% *p* < 0.05, Figure 1A,B, respectively). On the other hand, the contents of S1P, SFA1P, and CER in CBD group remained unchanged compared to the control subjects (+32.2%; −13.5%; +8.6%; *p* > 0.05; Figure 1C,D, respectively).

The simultaneous administration of the high-fat diet and CBD resulted in a reduction in the SFO, SFA, S1P, and SFA1P levels compared to the control group (−84.7%, −67.2%; −100.6%, −25.1%; *p* < 0.05; Figure 1A,B,C,D, respectively). Furthermore, the levels of SFA, S1P, SFA1P, and CER were markedly reduced (−137.7%; −301.2%; −68.9%; −42.2%; *p* < 0.05; Figure 1B,C,D,E, respectively), and the SFO level was significantly elevated compared to the HFD group (+68%; *p* < 0.05; Figure 1A).

### 2.2. Influence of CBD on the Sphingolipid Metabolism Enzymes in SAT and VAT of Rats Fed with Standard and High-Fat Diets

A Western blotting analysis revealed that in SAT, the expression of serine palmitoyltransferase, long-chain base subunit 1 (SPTLC1) was significantly decreased in all of the experimental groups compared to the control group (HFD: −32.2%; CBD: −65.4%; HFD+CBD: −76.6%; *p* < 0.05; Figure 2A). In VAT, we only observed a significant decline in SPTLC1 expression in the HFD+CBD group in comparison with that of the HFD-fed subjects (−30.5%; *p* < 0.05; Figure 2A). The induction of obesity with the high-fat diet resulted in an increase in SPTLC2 expression in both subcutaneous (+203%) and visceral (+76.9%) adipose tissue compared to that of the control (*p* < 0.05) (Figure 2B). Moreover, the chronic presence of CBD during the high-fat feeding caused a significant decrease in SPTLC2 expression in SAT (−209.3%) and VAT (−72.5%) compared with that in the group of rats fed with only a high-fat diet (*p* < 0.05) (Figure 2B).

As shown in Figure 3A, the CerS2 expression was significantly elevated only in the SAT in the group of HFD-fed rats compared to rats fed a standard chow (+47.2%; *p* < 0.05). In VAT, the CerS2 expression was reduced in the HFD+CBD group in comparison with that of the HFD-fed rats (−27.9%, *p* < 0.05).

On the other hand, we noticed a substantial increase in CerS4 expression in all of the experimental groups in comparison with the control subjects (HFD: +139.5%; CBD: +135.5%; HFD+CBD: 139.6%; *p* < 0.05; Figure 3B). In visceral adipose tissue, we exhibited a significant increase in CerS4 expression in both groups of rats fed with a high-fat chow and untreated or treated with CBD (+53.5% and +73.4%, respectively; *p* < 0.05; Figure 3B).

In the case of CerS5, we noticed that only high-fat chow caused a marked elevation of this protein’s expression, which was visible in the two types of adipose tissues (VAT: +26.2%; SAT: +34.1%; *p* < 0.05%; Figure 3C). Moreover, in VAT, this increase was significantly diminished by CBD injections compared to that in the HFD group (−27.4%, *p* < 0.05; Figure 3C). 

The total expression of CerS6 in SAT increased considerably compared to that of the control subjects in the case of animals fed with a high-fat chow and that of the animals that were injected with CBD (+78.3% and +72.5%, respectively; *p* < 0.05; Figure 3D). The simultaneous administration of the HFD and CBD caused a substantial decline in the CerS6 expression in comparison with that in the SAT of the HFD-fed rats (−84.3%, *p* < 0.05; Figure 3D). Interestingly, the total CerS6 expression in VAT was elevated prominently only in the HFD+CBD group compared to the rats fed with a standard or high-fat diet (+55%, *p* < 0.05 vs. control group; +63.4%, *p* < 0.05 vs. HFD group; Figure 3D).

The expression of enzymes involved in ceramide metabolism is presented in Figure 4. In subcutaneous adipose tissue, we observed a significant increase in the total ASAH1 expression (+58.6%, *p* < 0.05; Figure 4A) in the HFD group compared to the rats with fed a standard chow, and it further declined during the two weeks with CBD injections (−63%, *p* < 0.05; Figure 4A). In VAT, the total ASAH1 expression was unchanged in all of the examined groups (*p* > 0.05; Figure 4A). Similarly, significant alterations in ASAH2 expression were visible only in SAT. We noticed that there was pronounced increase in this protein’s expression after the HFD treatment, as well as the HFD and CBD treatment, compared to that in the control group (+45.1% and +59%, respectively, *p* < 0.05; Figure 4B).

The total SPHK1 expression did not change in subcutaneous adipose tissue. On the other hand, in visceral adipose tissue, we revealed a marked increase in the SPHK1 expression in the HFD-fed rats and in the rats treated by CBD (+76.9% and +37.5%, respectively, *p* < 0.05; Figure 4C). Importantly, we also observed that the two-week administration of CBD in the HFD rats resulted in a visible decrease in the SPHK1 expression (−65.6%, *p* < 0.05; Figure 4C). The SPHK2 expression level remained unchanged in both the SAT and VAT of all of the examined groups (Figure 4D).

### 2.3. Influence of CBD on the Expression of Insulin Signaling Pathway Proteins in SAT and VAT of Rats Fed with Standard and High-Fat Diets

In subcutaneous adipose tissue, there was a significant diminution in the phosphorylated GSK3β to unphosphorylated GSK3β ratio (−74.3%, *p* < 0.05; Figure 5B) for the high-fat diet, and there was a substantial elevation of the GSK3β phosphorylation ratio in the chronic presence of CBD compared to the control animals (+121.9%, *p* < 0.05; Figure 5B).

In visceral adipose tissue, the high-fat chow alone induced a decrease in the Akt phosphorylation ratio (−48.8%; *p* < 0.05; Figure 5A) compared to the control. However, while administering the high-fat diet, the administration of CBD had the opposite effect of elevating the pAkt to Akt ratio in comparison with those of the control (+47.3%, *p* < 0.05; Figure 5A) and HFD groups (+96.4%, *p* < 0.05; Figure 5A). Similarly, we found a reduced pGSK3β to GSK3β ratio in the rats fed a high-fat diet (35.5%, *p* < 0.05; Figure 5A) compared to that in the control group, and it was further considerably increased with the CBD injections.

## 3. Discussion

Over the last decade, phytocannabinoids, including CBD, have become emerging players in ameliorating the symptoms of insulin resistance, T2DM, and metabolic syndrome [26]. The most common features of the above-mentioned metabolic diseases are disturbances in lipid metabolism and subsequent insulin resistance. This problem is even more alarming because effective treatment of such diseases does not exist.

The effects of CBD on adipocyte lipid metabolism were first explored by Silvestri et al. in an in vitro study, demonstrating that CBD applied simultaneously with oleic acid (OA) reduced triacylglycerol accumulation in 3T3-L1 adipocytes [21]. Because the deposited TAGs may be esterified into various fractions, we resolved to evaluate the influence of CBD on a class of lipids that are more biologically active compared with other lipid fractions, namely, sphingolipids. Thus, in our experiment, which was performed on a rat model, the effects of CBD on sphingolipid concentration and their metabolism, e.g., for ceramide biosynthesis pathway enzymes were assessed. Furthermore, we formulated a hypothesis that CBD, by interfering with the sphingolipid metabolism pathway, may modulate adipocyte insulin sensitivity. Our previous study showed that chronic CBD treatment during excessive fat intake led to the alleviation of de novo CER synthesis, restored content of S1P, and an enhanced insulin signaling pathway and glycogen recovery in rat skeletal muscles [27].

The sphingolipid metabolic pathway includes simple molecules, such as ceramide, and a plurality of more complex sphingolipids, in which ceramide represents the central node in the biosynthesis and catabolism of sphingolipids [28]. The main route of ceramide formation is the de novo synthesis pathway, where the condensation of serine and palmitoyl-CoA by serine palmitoyl transferases (SPTLC1, SPTLC2) takes place [24]. It is well known that nutrient oversupply specifically, with saturated fatty acids intensifies the de novo ceramide synthesis pathway, which generates increased levels of CER in several tissues, including adipose tissue [17,29]. This phenomenon was also demonstrated in the present study in both types of adipose tissue derived from rats fed with a high-fat diet. Importantly, the higher CER level induced by the HFD was reduced in response to CBD injections. Interestingly, the decreased CER content in fat deposits after CBD administration was in line with the study conducted by Łuczaj et al., who indicated a decrease in CER levels in keratinocytes after topical application of CBD in UVA-irradiated rats [30].

The beneficial impact of CBD on subcutaneous adipose tissue observed in our experiment on insulin-resistant rats was mainly achieved through the inhibition of de novo ceramide synthesis, which manifested as a considerable decrease in the ceramide concentration of the precursor sphinganine (SFA). Moreover, this is consistent with the decreased expression of enzymes involved in sphingolipid synthesis, SPTLC2 and CerS6, after the CBD and HFD treatment. It is known that high CerS6 expression in the adipose tissue of obese humans is essential for the formation of unfavorable C16:0 ceramide, which is correlated with the development of insulin resistance in adipocytes, but not the whole body [18,31,32]. As we showed, CBD administration during the HFD feeding attenuated the CerS6 expression induced by lipid oversupply, which strongly suggests its positive metabolic effect. It is worth emphasizing that, in VAT, the inhibition of the de novo ceramide synthesis route was much more pronounced than in subcutaneous fat deposits. In this type of tissue, the reduction in SFA concentration was more visible, and a decreased expression of key de novo ceramide synthesis enzymes SPTLC1, SPTLC2, and CerS2/5 was observed. It is worth highlighting that CBD may act similarly to myriocin, a selective inhibitor of SPTLC that, through the inhibition of ceramide biosynthesis, induces profound changes in skeletal muscle ceramide content with an improvement in systemic insulin action [33]. However, due to the fact that myriocin exerts adverse effects, such as hepatotoxicity [34], CBD appears to be a more beneficial alternative in the treatment of IR.

In addition to the de novo synthesis and sphingomyelin hydrolysis, CER can also be produced from the breakdown of sphingosine through the action of ceramide synthases (CerS1-6) in the salvage pathway, which is even responsible for 90% of total sphingolipid biosynthesis [28,35]. Once formed, CER can undergo degradation because of the actions of ceramidases (ASAH1, ASAH2) toward sphingosine, which can return to the sphingolipid pathway or be transformed by sphingosine kinases (SPHK1, SPHK2) into sphingosine-1 phosphate (S1P) [36,37]. In SAT, the substantial decrease in the intracellular SFO content with the simultaneously lowered expression of enzymes involved in the salvage pathway (CerS6) and CER catabolism (ASAH1) proved the inhibition of both routes after CBD injections in IR rats. On the other hand, in VAT, under conditions of high fatty acid availability, cannabidiol favored ceramide catabolism more than its synthesis in the salvage pathway, which was reflected by the higher SFO level with increased CerS6 and ASAH1 expression. This effect seems to be very important because SFO, as a ceramide derivative with proapoptotic properties, modulates key cellular processes, has thereby been linked with metabolic disorders [38,39]. However, a further step of sphingolipid catabolism in VAT was inhibited, as evidenced by the lower SPHK1 expression with a diminished intracellular S1P content. We hypothesized that the pronounced decline in S1P concentration may also have resulted from the intensified transport of this molecule outside the cells or the redirection of S1P to further degradation into hexadecenal and ethanolamide-1-phosphate. However, at this stage of research, these are just assumptions that should be clarified in the future. Our findings are in accordance with those of the study conducted by Wang et al., where, in mice with diet-induced obesity, SPHK1 deficiency was associated with reduced S1P concentration and enhanced insulin sensitivity in epididymal adipose tissue [40]. Moreover, numerous studies have shown that obese subjects had increased SPHK1 expression accompanied by raised deposition of S1P not only in adipose tissue, but also several other tissues [40,41]. Although the specific role played by S1P in metabolic maladies remains unclear [32], some authors suggested that genetic disturbance of the SPHK1/S1P signaling pathway in HFD-fed mice led to improved systemic insulin sensitivity [40]. Based on this, we suspect that inhibition of SPHK1 by CBD may be considered as a potential therapeutic approach to restoring normal insulin activity in obesity.

Among all of the sphingolipids, there is a particular interest in ceramide because it promotes the development of adipocyte insulin resistance to the greatest extent [42]. Ceramide disrupts transport of glucose into the cells through the dephosphorylation of protein kinase B (Akt/PKB), by blocking the transfer of Akt/PKB to the plasma membrane, or by targeting protein phosphatase 2A (PP2A). Moreover, accumulation ceramide in VAT more than in SAT is associated with obesity and metabolic syndrome [43,44]. Many recent publications have summarized that the diminution of ceramide deposition in adipocytes improved their insulin sensitivity. As a consequence, it contributed to the normalization of glucose homeostasis in the whole body, which also resolved the progression of hepatic steatosis [45,46,47]. In the present study, CBD administration in HFD-fed rats elevated the pAkt (Ser473)/Akt and pGSK3β (Ser9)/GSK3β expression ratios, but only in VAT. These findings corresponded with the reduced ceramide content, which proved the restoration of insulin sensitivity through the action of the examined phytocannabinoid in visceral fat deposits. However, we did not observe any crucial changes in the expression of insulin pathway proteins in SAT between the groups of rats that were fed with a high-fat chow and were untreated or treated with CBD. In this fat deposit, the only change that was observed was an increase in the pGSK3β (Ser9)/GSK3β ratio and the trend toward an increase in the p/Akt (Ser473)/Akt ratio only after CBD treatment. This may have resulted from the reduced lipolytic activity and higher insulin sensitivity of the subcutaneous fat regions, which are specialized to provide long-term storage of excessive fatty acids, thus protecting other tissues from lipotoxic effects [8,48]. These CBD-dependent changes in insulin signaling pathway are consistent with our previous study where two-week CBD administration caused a substantial reduction in insulin plasma concentration and the trend toward a decrease in HOMA-IR value in the HFD-fed rats [27].

Our observations suggest the distinct effects of CBD on two types of fat deposits in the context of intracellular insulin signaling, and they complement the reports on the anti-diabetic properties of CBD [49,50].

## 4. Materials and Methods

### 4.1. Animal Model

This study was conducted on male Wistar rats (initial body weight: 70–100 g), which were housed in approved animal-holding facilities (at 22 ± 2 °C, a 12 h/12 h light–dark cycle, with unlimited access to water and to a selected chow). Experiments were carried out after one week of the animals’ acclimatization. In our experiment, we used n = 10 animals in every experimental group, as was counted using an internet service (http://www.biomath.info, accessed on 1 January 2021). The animals were randomly selected for particular groups, and their body mass was monitored during the study. To make sure that the animals did not compensate by eating and drinking less, the amounts of food and water were monitored. This study was approved by the Animal Ethics Committee in Olsztyn (approval number 71/2018).

### 4.2. Study Design

The control group and CBD group were fed with standard rat chow (kcal distribution = 62% carbohydrates, 16% fat, and 22% protein). In the HFD and HFD+CBD groups, insulin resistance was induced through the administration of a high-fat diet (HFD; kcal distribution = 20% carbohydrates, 59% fat, and 21% protein; Labofeed B, Kcynia, Poland) for a period of 7 weeks in an amount corresponding to the daily caloric intake of the animals receiving standard chow. The synthetic CBD (THC Pharm GmbH, Frankfurt, Germany) was administered intraperitoneally at a concentration of 10 mg/kg of body mass once a day and, at the same time, for 14 consecutive days (sixth and seventh weeks of the experiment). The dose of CBD was chosen based on an analysis of data from the literature [51]. Immediately before administration, CBD was dissolved in the vehicle (3:1:16, namely ethanol, Tween-80, and 0.9% NaCl, respectively). The rats were anesthetized through the inhalation of isoflurane, and then subcutaneous and visceral fat samples were taken from the sleeping animals. The collected tissues were frozen at the temperature of liquid nitrogen (–196 °C) and stored for further analysis at −80 °C.

### 4.3. Sphingolipid Analysis

The ceramide, sphinganine (SFA), sphingosine (SFO), sphinganine-1-phosphate (SFA1P), and sphingosine-1-phosphate (S1P) contents in the visceral and subcutaneous adipose tissues were estimated by using high-performance liquid chromatography (HPLC), as described in the method developed by Min et al. and as previously described [52]. In brief, the lipids from the homogenates of rat adipose tissue were extracted with the use of chloroform. Subsequently, an aliquot of the lipid extract was transferred into a fresh tube containing an internal standard (N-palmitayl-D-erythrospingosine). Then, the samples were subjected to alkaline hydrolysis to deacylate the ceramide. Two sphingolipid metabolites that were released from ceramide during the analytical process, free sphinganine and sphingosine, were then converted into their o-phthalaldehyde derivatives and analyzed by using a standard HPLC system (PROSTAR; Varian Inc., Palo Alto, CA, USA) equipped with a fluorescence detector and C-18 reversed-phase columns (Varian Inc. Omnispher 5, 4, 6 × 150 mm).

### 4.4. Western Blotting Analysis

The expression of proteins that were directly engaged in sphingolipid metabolism and the insulin signaling pathway was detected by using a routine Western blotting procedure, as previously described [53]. In brief, samples of the visceral and subcutaneous adipose tissues were homogenized in RIPA buffer with the addition of protease and phosphatase inhibitors (Roche). Next, the total protein concentration in the homogenates of the adipose tissues was estimated with a bicinchoninic acid assay (BCA) using bovine serum albumin (BSA) as a standard. After the dilution of the homogenates, the same amount of protein (20 μg) was loaded and separated using 10% Criterion TGX Stain-Free precast gel electrophoresis and shifted into nitrocellulose membranes. Afterwards, the membranes were blocked with TTBS buffer containing 5% nonfat dry milk or BSA for phosphorylated proteins for 90 min at room temperature. Subsequently, membranes were immunoblotted with primary antibodies: serine palmitoyltransferases (SPTLC1, SPTLC2; 1:500; Abcam, Cambridge, UK), ceramide synthases (CerS2, CerS4, CerS5, CerS6; 1:500; Thermo Scientific, Rockford, IL, USA), sphingosine kinases (SPHK1, SPHK2; 1:500; Sigma Aldrich, St. Louis, MO, USA), acid ceramidase (ASAH1, 1:500; Santa Cruz Biotechnology, Paso Robles, CA, USA), neutral ceramidase (ASAH2, 1:500; Santa Cruz Biotechnology, Paso Robles, CA, USA), protein kinase B (Akt/PKB, 1:500; Cell Signaling Technology, Danvers, MA, USA), phosphorylated protein kinase B (pAkt/PKB (Ser473), 1:500; Cell Signaling Technology, Danvers, MA, USA), glycogen synthase kinase 3ß (GSK3ß 1:500, Santa Cruz Biotechnology, Paso Robles, CA, USA), and phosphorylated glycogen synthase kinase 3ß (p/GSK3ß (Ser9) 1:500, Santa Cruz Paso Robles, CA, USA). Then, the nitrocellulose membranes were incubated with the appropriate horseradish-peroxidase-labeled (HRP) secondary anti-goat or anti-mouse antibodies. The protein expressions were assayed densitometrically using the ChemiDoc visualization system (Image Laboratory Software Version 6.0.1; BioRad, Warsaw, Poland). Equal protein loading was presented by CCD imager and UV illumination using TGX stain-free system. Next, the image of total protein and protein of interest overlapped in ImageLab system. The expression of all of the proteins was standardized to the total protein expression by using the stain-free method, and the control was set to 100%.

### 4.5. Statistical Analysis

All data were expressed as the mean and standard deviation. The assumptions of the methods used in our analysis, that is, the normality of the data distribution (Shapiro–Wilk test) and homogeneity of the variance (Bartlett’s test), were tested. Statistical differences were established based on the results of one-way ANOVA followed by an appropriate post hoc test (i.e., pairwise Student t-test) using GraphPad Prism 7. Differences in the levels of analyzed parameters were considered at *p* < 0.05.

## 5. Conclusions

Our data indicated that, under conditions of overnutrition, CBD injections modulated sphingolipid metabolism, leading finally to a reduction in the ceramide content in two types of adipose tissues. In visceral adipose tissue, the action of CBD was more directed toward the inhibition of the de novo ceramide synthesis pathway and the intensification of its catabolism. On the other hand, in subcutaneous fat depots, CBD restrained the de novo ceramide synthesis and salvage synthesis routes. All of these changes ultimately sensitized adipose tissue to insulin, especially in the visceral deposits. Therefore, CBD use may be considered as a potential therapeutic strategy for treating or reducing insulin resistance, T2DM, and metabolic syndrome.

## Figures and Tables

**Figure 1 ijms-23-05382-f001:**
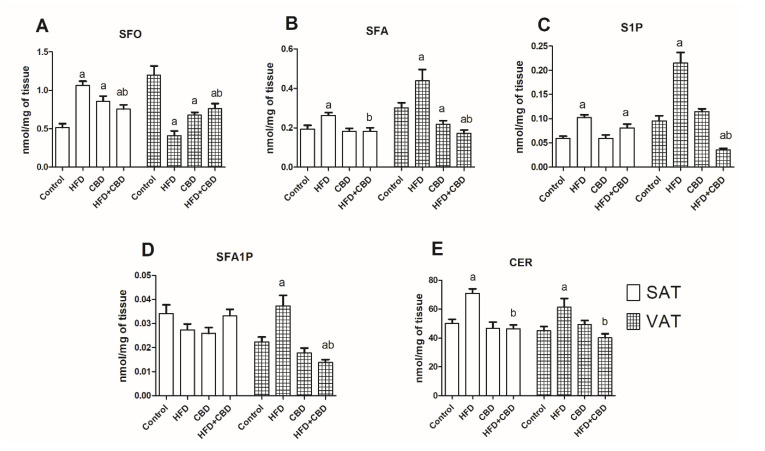
The concentrations of sphingolipids in the subcutaneous (SAT) and visceral (VAT) adipose tissues of rats fed a standard rat chow (control) or high-fat diet (HFD) after chronic cannabidiol (CBD) treatment: (**A**) sphingosine (SFO), (**B**) sphinganine (SFA), (**C**) sphingosine-1-phosphate (S1P), (**D**) sphinganine-1-phosphate (SFA1P), (**E**) ceramide (CER). The data are presented as mean values ± SD, n = 10 in each group. ^a^
*p* < 0.05: significant difference: control group vs. experimental group; ^b^
*p* < 0.05: significant difference: HFD vs. HFD+CBD.

**Figure 2 ijms-23-05382-f002:**
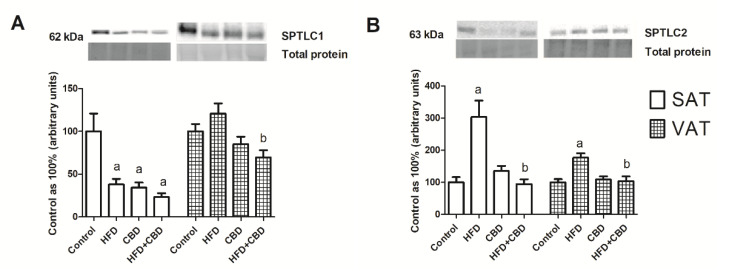
The total expression of proteins from de novo sphingolipid synthesis, e.g., (**A**) serine palmitoyltransferase, long-chain base subunit 1 (SPTLC1) and (**B**) serine palmitoyltransferase, long-chain base subunit 2 (SPTLC2), in the subcutaneous (SAT) and visceral (VAT) adipose tissues of rats fed a standard diet (control) or high-fat diet (HFD) after chronic CBD treatment. The total expression of the above proteins was standardized to the total protein expression, and the control group was set to 100%. The data are expressed as mean values ± SD, n = 6 in each group. ^a^
*p* < 0.05: significant difference: control group vs. experimental group; ^b^
*p* < 0.05: significant difference: HFD vs. HFD+CBD.

**Figure 3 ijms-23-05382-f003:**
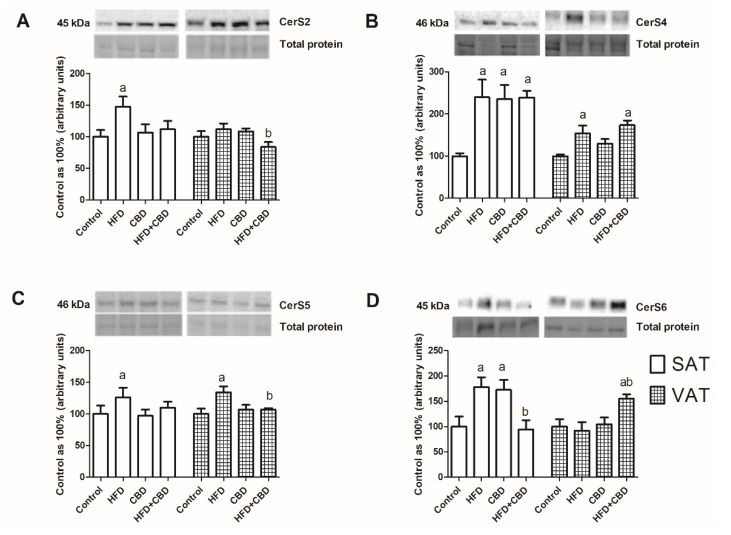
The total expression of proteins from both de novo ceramide synthesis and salvage pathways, e.g., (**A**) ceramide synthase 2 (CerS2), (**B**) ceramide synthase 4 (CerS4), (**C**) ceramide synthase 5 (CerS5), and (**D**) ceramide synthase 6 (CerS6), in the subcutaneous (SAT) and visceral (VAT) adipose tissues of rats fed a standard diet (control) or high-fat diet (HFD) after CBD treatment. The total expression of the above proteins was standardized to the total protein expression, and the control group was set to 100%. The data are expressed as mean values ± SD, n = 6 in each group. ^a^
*p* < 0.05: significant difference: control group vs. experimental group; ^b^
*p* < 0.05: significant difference: HFD vs. HFD+CBD.

**Figure 4 ijms-23-05382-f004:**
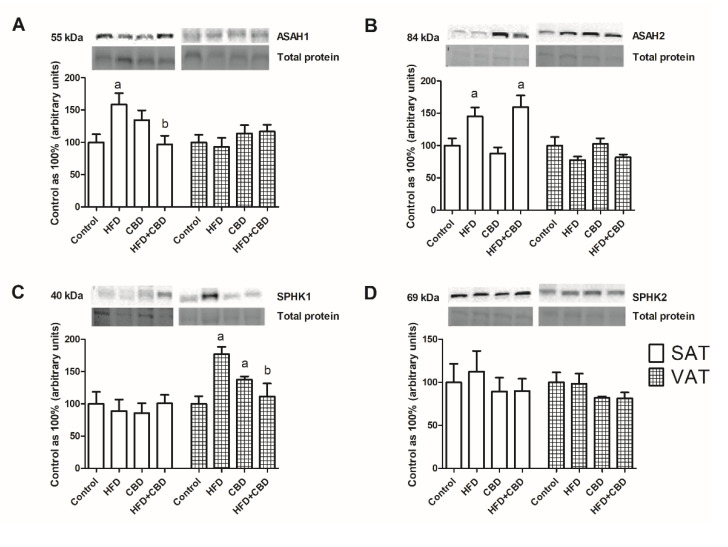
The total expression of proteins from sphingolipid catabolism, e.g., (**A**) acid ceramidase (ASAH1), (**B**) acid ceramidase (ASAH2), (**C**) sphingosine kinase 1 (SPHK1), and (**D**) sphingosine kinase 2 (SPHK2), in the subcutaneous (SAT) and visceral (VAT) adipose tissues of rats fed a standard diet (control) or high-fat diet (HFD) after chronic CBD treatment. The total expression of the above proteins was standardized to the total protein expression, and the control group was set to 100%. The data are expressed as mean values ± SD, n = 6 in each group. ^a^
*p* < 0.05: significant difference: control group vs. experimental group; ^b^
*p* < 0.05: significant difference: HFD vs. HFD+CBD.

**Figure 5 ijms-23-05382-f005:**
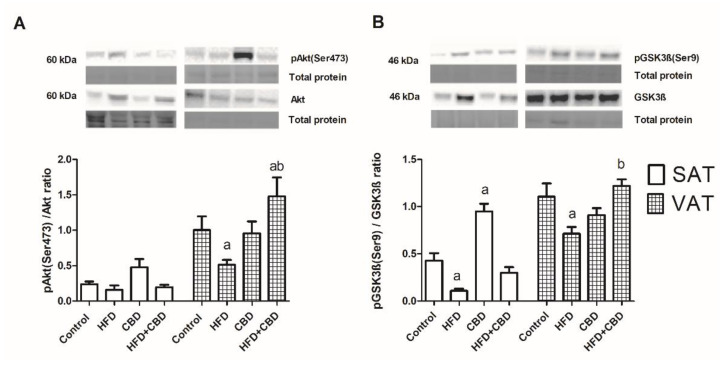
The total expression of proteins belonging to the insulin signaling pathway, e.g., (**A**) pAkt (Ser 473)/Akt and (**B**) pGSK3ß (Ser9)/GSK3ß, in the subcutaneous (SAT) and visceral (VAT) adipose tissues of rats fed a standard diet (control) or high-fat diet (HFD) after chronic cannabidiol (CBD) treatment. The total expression of the above proteins was standardized to the total protein expression, and the control group was set to 100%. The data are expressed as mean values ± SD, n = 6 in each group. ^a^
*p* < 0.05: significant difference: control group vs. experimental group; ^b^
*p* < 0.05: significant difference: HFD vs. HFD+CBD.

## Data Availability

The data presented in this study are available on request from the corresponding author.

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
