# Peer review of "Distinct Effects of Cannabidiol on Sphingolipid Metabolism in Subcutaneous and Visceral Adipose Tissues Derived from High-Fat-Diet-Fed Male Wistar Rats"

_ijms, 2022, doi:10.3390/ijms23105382_

Round 1

Reviewer 1 Report

The manuscript shows some interesting data on the effect of CBD on sphingolipid metabolism in Subcutaneous and Visceral Adipose Tissues.  The manuscript is in general well written. Some Scientific English styling is needed. There are some issues that need to be addressed before the manuscript can be accepted for publication as stated below.

Comments:

Line 13 in Abstract: Instead of “We indicated” I would suggest to write “We show”.

Line 29-30 in Introduction: There are some wordings in small letters that seems not to be in the right place.

Line 38-42 – The sentence is too long and there is a syntactic problem. Please divide the sentence into two.

Line 59: 9 in Δ9-THC should be in superscript.

Line 116: Please write the full name of the enzyme SPTLC1 (first time mentioned).

Line 124: A paranthesis is lacking after SAT.

Line 127: I would suggest to write ”As shown in figure 3A”. (In the current writing, it sounds that you already have shown the data previously…).

The total protein analyses in Figure 2  and in the other figures are not convincing. Can a control antibody such as α-Tubulin or β-Actin be used? I think this should also be done for the other Western blot analyses. In the Supplementary data of the original images, please indicate which bands were used for the total protein in the manuscript.

Figure 5: It seems that there is a problem with the calculation of the band intensities. The p-Akt is high in CBD treated group and in the graph the authors present the data that there is no change.  The Densitrometry should be redone with accuracies. I believe this should also be done for the other fi

The manuscript shows some interesting data on the effect of CBD on sphingolipid metabolism in Subcutaneous and Visceral Adipose Tissues.  The manuscript is in general well written. Some Scientific English styling is needed. There are some issues that need to be addressed before the manuscript can be accepted for publication as stated below.

Comments:

Line 13 in Abstract: Instead of “We indicated” I would suggest to write “We show”.

Line 29-30 in Introduction: There are some wordings in small letters that seems not to be in the right place.

Line 38-42 – The sentence is too long and there is a syntactic problem. Please divide the sentence into two.

Line 59: 9 in Δ9-THC should be in superscript.

Line 116: Please write the full name of the enzyme SPTLC1 (first time mentioned).

Line 124: A paranthesis is lacking after SAT.

Line 127: I would suggest to write ”As shown in figure 3A”. (In the current writing, it sounds that you already have shown the data previously…).

The total protein analyses in Figure 2  and in the other figures are not convincing. Can a control antibody such as α-Tubulin or β-Actin be used? I think this should also be done for the other Western blot analyses. In the Supplementary data of the original images, please indicate which bands were used for the total protein in the manuscript.

Figure 5: It seems that there is a problem with the calculation of the band intensities. The p-Akt is high in CBD treated group and in the graph the authors present the data that there is no change.  The Densitrometry should be redone with accuracies. I believe this should also be done for the other figures too.

A conclusive figure ought to be made showing the effect of CBD+/- HFD on the various parameters analyzed in the manuscript.

All references lack “issue number” and “page numbers”.

gures too.

A conclusive figure ought to be made showing the effect of CBD+/- HFD on the various parameters analyzed in the manuscript.

All references lack “issue number” and “page numbers”.

Author Response

Dear Sir or Madam,

Please find enclosed our revised manuscript entitled " Distinct Effects of Cannabidiol on Sphingolipid Metabolism in Subcutaneous and Visceral Adipose Tissues Derived from High-Fat-Diet-Fed Male Wistar Rats." (authors: Klaudia Berk, Karolina Konstantynowicz-Nowicka, Tomasz Charytoniuk, Ewa Harasim-Symbor, Adrian Chabowski )

Below, based on suggestions, we present all the changes that we made in our article, which was improved and reorganized completely. Furthermore, the whole manuscript was significantly corrected and improved grammatically

Reviewer 1

The manuscript shows some interesting data on the effect of CBD on sphingolipid metabolism in Subcutaneous and Visceral Adipose Tissues. The manuscript is in general well written. Some Scientific English styling is needed. There are some issues that need to be addressed before the manuscript can be accepted for publication as stated below.

Comments:

Line 13 in Abstract: Instead of “We indicated” I would suggest to write “We show”.
Authors: The sentence has been changed (line 14)

Line 29-30 in Introduction: There are some wordings in small letters that seems not to be in the right place.
Authors: The sentence has been changed (line 31)

Line 38-42 – The sentence is too long and there is a syntactic problem. Please divide the sentence into two.

Authors: The sentence has been changed: On the other hand, preferential accumulation of fat in subcutaneous deposits, such as those detected in most women, positively correlates with metabolically healthy obesity. It is a condition in which, despite the occurrence of obesity, individuals may be protected from T2DM and the development of cardiovascular diseases (lines: 39-43)

Line 59: 9 in Δ9-THC should be in superscript.
Authors: The sentence has been changed (line 63)

Line 116: Please write the full name of the enzyme SPTLC1 (first time mentioned). (lines:
Authors: The full name has been added (line 120-121)

Line 124: A paranthesis is lacking after SAT.

Authors: The symbol has been added. (line 129)

Line 127: I would suggest to write ”As shown in figure 3A”. (In the current writing, it sounds that you already have shown the data previously…).

Authors: The sentence has been changed. (line 131)

The total protein analyses in Figure 2  and in the other figures are not convincing. Can a control antibody such as α-Tubulin or β-Actin be used? I think this should also be done for the other Western blot analyses. In the Supplementary data of the original images, please indicate which bands were used for the total protein in the manuscript.

Authors: There are two methods of Western blot normalization. One is the use of housekeeping protein as was suggested by the Reviewer. However, recent studies indicated that expression of housekeeping proteins is not stable and may be altered by some experimental conditions. Thus, in our experiment we selected other normalization method namely normalization by sum using total protein staining. The lower panel is the image of Stain-Free labeled proteins after membrane transfer that is used for normalization. The Stain-Free system from Bio-Rad that was used in our study takes chemical labeling of the sample proteins. The labeling moiety, a trihalo compound, is incorporated into the proprietary pre-cast gel. When this gel is exposed to UV light, the trihalo label covalently binds tryptophan residues in the sample proteins and forms a cross-linked fluorescent product. Fluorescently-modified proteins are then detected with a CCD imager and UV illumination. Next, the image of total protein and protein of interest overlap in ImageLab system that is attached to ChemiDoc visualization system. Total protein staining after membrane transfer is possibly the most reliable and accurate method for normalization of Western blot data:

Analytical Biochemistry, Volume 440, Issue 2, 15 September 2013, Pages 186-188: Stain-Free total protein staining is a superior loading control to β-actin for Western blots. Jennifer E.Gilda, Aldrin V.Gomes.

Figure 5: It seems that there is a problem with the calculation of the band intensities. The p-Akt is high in CBD treated group and in the graph the authors present the data that there is no change.  The Densitrometry should be redone with accuracies. I believe this should also be done for the other fi

Authors: We understand the Editors’s concern because the obtained data after normalization to total protein sometimes may be changed compared to data observed in the chemiluminescent blot image. Thus, in the figures as well as in the supplementary data we presented not only chemiluminescent blot  but also the stain-free blot image. 

We agree with the Editor that Western blot procedure lacs some important information. Thus, we added more precise description into Materials and methods section how normalization was conducted (lines: 403-405)

A conclusive figure ought to be made showing the effect of CBD+/- HFD on the various parameters analyzed in the manuscript.

Authors: The graphical abstract has been added.

All references lack “issue number” and “page numbers”.

Authors: The references has been changed.

We believe that the manuscript is now more comprehensive and suitable for publication in the International Journal of Molecular Sciences

Yours faithfully,

Klaudia Berk

Department of Physiology,

Medical University of Bialystok,

15-222 Bialystok, Mickiewicz Str. 2C, Poland

Reviewer 2 Report

  1. One of these comorbidities is insulin resistance (IR), which is a pathophysiological state where vital metabolic organs (liver, muscle, adipose tissues) demonstrate reduced sensitivity to the glucose-lowering activity of insulin Insulin action and resistance in obesity and type 2 diabetes. This sentence has some problems and needs to be checked and corrected. Also, citing the latest literature about insulin resistance will be helpful.

https://doi.org/10.1016/j.ijbiomac.2022.03.004

https://doi.org/10.1021/acsomega.1c00631

  1. On the other hand, preferential accumulation of fat in subcutaneous deposits, such as those detected in most women, positively correlates with metabolically healthy obesity, a condition in which, despite the occurrence of obesity, individuals may be protected from type 2 diabetes and the development of cardiovascular diseases. Erratic use of abbreviations. I found type 2 diabetes abbreviated in some places in the manuscript, while the full word is used in other places. The authors are advised to cross-check the manuscript for proper abbreviation usage.
  2. d affects lipid metabolism, but, unlike Δ9-THC, it lacks potential for abuse. Δ9-THC. What??
  3. All the results are nicely compiled and presented.
  4. The animals were randomly selected for particular groups, and their body mass was monitored during the study. No prior weight filter or sex filter was used?
  5. A graphical abstract depicting the whole summary of the work is advised.

Author Response

Dear Sir or Madam,

Please find enclosed our revised manuscript entitled " Distinct Effects of Cannabidiol on Sphingolipid Metabolism in Subcutaneous and Visceral Adipose Tissues Derived from High-Fat-Diet-Fed Male Wistar Rats." (authors: Klaudia Berk, Karolina Konstantynowicz-Nowicka, Tomasz Charytoniuk, Ewa Harasim-Symbor, Adrian Chabowski )

Below, based on suggestions, we present all the changes that we made in our article, which was improved and reorganized completely. Furthermore, the whole manuscript was significantly corrected and improved grammatically

Reviewer 2

One of these comorbidities is insulin resistance (IR), which is a pathophysiological state where vital metabolic organs (liver, muscle, adipose tissues) demonstrate reduced sensitivity to the glucose-lowering activity of insulin Insulin action and resistance in obesity and type 2 diabetes. This sentence has some problems and needs to be checked and corrected. Also, citing the latest literature about insulin resistance will be helpful.

https://doi.org/10.1016/j.ijbiomac.2022.03.004

https://doi.org/10.1021/acsomega.1c00631

Authors: The sentence has been changed into : „One of these comorbidities is insulin resistance (IR), which is a pathophysiological state where vital metabolic organs (liver, muscle, adipose tissues) demonstrate reduced sensitivity to the activity of insulin.” The references has been added (number 2 and 3)

  1. On the other hand, preferential accumulation of fat in subcutaneous deposits, such as those detected in most women, positively correlates with metabolically healthy obesity, a condition in which, despite the occurrence of obesity, individuals may be protected from type 2 diabetes and the development of cardiovascular diseases.
    Erratic use of abbreviations. I found type 2 diabetes abbreviated in some places in the manuscript, while the full word is used in other places. The authors are advised to cross-check the manuscript for proper abbreviation usage.

    Authors: the nomenclature has been unified.

  2. d affects lipid metabolism, but, unlike Δ9-THC, it lacks potential for abuse. Δ9-THC. What??

    Authors: The sentence has been changed. it lacks abuse potential (line 63-64)

  3. The animals were randomly selected for particular groups, and their body mass was monitored during the study. No prior weight filter or sex filter was used?
    Authors: We performed this study on male Wistar rats only with initial body weight: 70–100 g

  4. A graphical abstract depicting the whole summary of the work is advised.

    Authors: The graphical abstract has been added.

We believe that the manuscript is now more comprehensive and suitable for publication in the International Journal of Molecular Sciences

Yours faithfully,

Klaudia Berk

Department of Physiology,

Medical University of Bialystok,

15-222 Bialystok, Mickiewicz Str. 2C, Poland

Round 2

Reviewer 1 Report

The manuscript has been improved and is suitable for publication.